# Development of a PVY Resistant Flue-Cured Tobacco Line via EMS Mutagenesis of *eIF4E*

**Lu Zhao** [1,2,3], **Wenzheng Li** [1,2,3], **Bingwu Wang** [1,2,3], **Yulong Gao** [1,2,3], **Xueyi Sui** [1,2,3],
**Yong Liu** [1,2,3], **Xuejun Chen** [1,2,3], **Xuefeng Yao** [4,5], **Fangchan Jiao** [1,2,3] **and Zhongbang Song** [1,2,3,*] 

[1] Tobacco Breeding and Biotechnology Research Center, Yunnan Academy of Tobacco Agricultural Sciences, Kunming 650021, China; zhaolu66@outlook.com (L.Z.); lwz67@163.com (W.L.); bwwang76@hotmail.com (B.W.); gyl3000@163.com (Y.G.); xueyisui2017@hotmail.com (X.S.); yong2liu@sina.com (Y.L.); cxjkm@163.com (X.C.); jfc99002@163.com (F.J.)
[2] Key Laboratory of Tobacco Biotechnological Breeding, Kunming 650021, China
[3] National Tobacco Genetic Engineering Research Center, Kunming 650021, China
[4] Key Laboratory of Plant Molecular Physiology, Institute of Botany, The Chinese Academy of Sciences, Beijing 100093, China; yxfun@163.com
[5] The University of Chinese Academy of Sciences, Beijing 100049, China
[*] Correspondence: zbsoon@vip.163.com; Tel.: +86-136-2877-1356

**Abstract:** Recessive resistance against potyviruses, such as *Potato virus Y* (PVY), relies on mutations in the eukaryotic translation initiation factor 4E (eIF4E) or one of its isoforms. The *eIF4E1-S* mutants of burley tobacco (*Nicotiana tabacum* L.) exhibit recessive resistance against PVY strains. Here, we developed a TILLING population of flue-cured tobacco (*N. tabacum* cv. Yunyan87) using ethyl methanesulfonate (EMS) to identify *eIF4E1-S* mutants. M3 plants homozygous for a nonsense mutation in exon 1 of the *eIF4E1-S* gene demonstrated resistance against PVY$^{MN}$. These M3 plants were backcrossed to 'Yunyan87', and BC$_4$F$_3$ plants were screened using derived cleaved amplified polymorphic sequence (dCAPS) markers. BC$_4$F$_3$ plants showing agronomic traits comparable to the recurrent parent 'Yunyan87' and resistance against PVY$^O$, PVY$^N$, and PVY$^{NTN}$ strains were identified. These genotypes would provide useful germplasm for future tobacco improvement and would aid in basic research on PVY resistance in flue-cured tobacco.

**Keywords:** *eIF4E1-S*; potyvirus strains; *Nicotiana tabacum* cv. Yunyan87; EMS-induced nonsense mutation; TILLING; dCAPS markers

## 1. Introduction

*Potato virus Y* (PVY) is the type member of the *Potyviridae* family, one of the largest genera of plant viruses [1]. PVY is transmitted by aphids and is one of the most destructive viral pathogens, infecting a wide range of plant species of the *Solanaceae* family, including potato (*Solanum tuberosum*), tomato (*Solanum lycopersicum*), pepper (*Capsicum annuum*), and tobacco (*Nicotiana tabacum*) [2]. PVY infection can cause severe symptoms, causing a significant reduction in crop yield and quality [3,4]. In potato, PVY has been reported to cause seed degeneration, resulting in up to 90% yield loss, depending on the PVY isolate [5]. In cultivated tobacco, PVY causes leaf necrosis and reduces leaf nicotine content, significantly reducing leaf yield (up to 100%) and leaf quality, respectively [6].

In tobacco, the *va* locus on chromosome 21, originally identified in the UV-mutagenized Virgin A Mutant (VAM), is the main source of PVY resistance [7,8]. VAM carries a 1 Mb deletion on chromosome 21, and VAM-type PVY resistance is conditioned by two separate recessive genes, *va1* and *va2*, responsible for limiting the cell-to-cell movement and accumulation of PVY, respectively [9]. Unlike

plants possessing the *va* locus, those containing the *VAM* locus confer potyvirus resistance in the presence of undesirable traits, such as small leaves, reduced yield [10], and poor aroma [11]. The PVY resistant tobacco plants carrying the *VAM* locus lack trichome exudates, presumably because the large deletion encompassing the *VAM* locus contains the *va* gene and probably other genes that control the morphology of tobacco leaves and biosynthesis of leaf surface compounds [8].

The potyviral genome-linked protein (VPg) interacts with the eukaryotic initiation factor 4E (eIF4E), 4G (eIF4G), and their isoforms [12–14] in host plants, as it mimics the 5'-cap structure of mRNA, thus promoting potyvirus multiplication in host cells [15]. Mutations in the *eIF4E* or *eIF(iso)4E* gene play an important role in recessive resistance against or reduced susceptibility to potyviruses. *Arabidopsis thaliana* plants containing mutations or disruptions in *eIF(iso)4E* genes exhibit resistance against different potyviruses, including *Turnip mosaic virus* (TuMV), *Lettuce mosaic virus* (LMV) [16], *Tobacco etch potyvirus* (TEV) [15], *Plum pox virus* (PPV) [17], and *Clover yellow vein virus* [18]. Mutations in *eIF4E* or *eIF(iso)4E* also confer resistance against potyviruses among *Solanaceae* species. For example, the *pvr2* locus in pepper, which corresponds to the *eIF4E* gene, confers recessive resistance against PVY [19]. In tomato, *pot-1*, an ortholog of the *eIF4E* gene, controls resistance to PVY [20]. Transcriptomic analysis of a tobacco recombinant inbred line (RIL) population has shown that *eIF4E* genes are responsible for susceptibility to potyviruses [21], while the absence of *eIF4E* isoforms in plants possessing *VAM* and *va* loci leads to recessive resistance against PVY [8,22]. Julio et al. [22] developed the ethyl methanesulfonate (EMS) mutant collection from seeds of a PVY susceptible burley tobacco line BB16NN and identified two mutations (W50* and W53*) resulting in premature stop codons in exon 1 of the *eIF4E1-S* gene. The results of enzyme-linked immunosorbent assay (ELISA) demonstrated that burley tobacco mutants carrying the W50* and W53* mutations were resistant to the PVY$^N$ isolate.

*Nicotiana tabacum* cv. Yunyan87 is the most widely cultivated flue-cured tobacco in China, accounting for more than 50% of the national acreage used for tobacco cultivation (through peer communication). Previously, flue-cured tobacco lines developed from the VAM line showed undesirable agronomic traits because of the large deletion on chromosome 21. In this study, we performed EMS mutagenesis to introduce point mutations in the first exon of the *eIF4E1-S* gene. Homozygous mutants of 'Yunyan87' carrying a nonsense mutation in exon 1 of the *eIF4E1-S* gene were selected from the M3 population by targeting induced local lesions in genomes (TILLING) [23] and backcrossed to 'Yunyan87'. The backcross progeny (BC$_4$F$_3$) was screened using derived cleaved amplified polymorphic sequence (dCAPS) markers, and lines showing comparable agronomic traits, such as 'Yunyan87', and resistance against various PVY strains were identified.

## 2. Materials and Methods

### 2.1. Plant Material and EMS (Ethyl Methane Sulfonate) Treatment

Seeds of *N. tabacum* L. cv. Yunyan87 were obtained from the Yunnan Academy of Tobacco Agricultural Sciences (Yunnan, China) and mutagenized with EMS (Sigma, St. Louis, MO, USA), as described previously [23]. Briefly, tobacco seeds were soaked in 50% detergent for 12 min, washed with sterile de-ionized water, and then soaked in de-ionized water for 12 h. To conduct the EMS treatment, sterilized tobacco seeds were immersed in a bottle containing 50 mL of 0.5% EMS for 12 h and then rinsed 6–8 times (1 min each time) with de-ionized water. The seed surface was dried by placing the seeds on a filter paper in the Büchner funnel under vacuum for 1 min. Seeds were then sown in floating trays, and seedlings were transplanted in the field, according to standard tobacco cultivation practices. Eight seeds from each M1 line were sown in the field, and the EMS-mutagenized M2 population with 1536 M2 individual plants was generated.

### 2.2. DNA Extraction and Sample Pooling

Young leaf materials were harvested from these M2 plants, transferred to 96-well plates containing two steel beads per well, and ground using a bead mill. Genomic DNA was isolated using DNeasy

96 Plant Kit (Qiagen, Germantown, MD, USA) and quantified on a 0.8% agarose gel. The final concentration of each DNA sample was adjusted to 40 ng/μL. The M2 library of 'Yunyan87' mutants was constructed by pooling the DNA samples 8-fold and loaded onto 96-well plates for use as a PCR template.

### 2.3. PCR Amplification

To identify EMS-induced mutations in the first exon of *eIF4E1-S* (accession number: Nitab4.5_0002814g0120.1), gene-specific primers, isoform_c_mF_1 and isoform_c_mR_1 (Supplementary Table S1), were designed to amplify a 489-bp fragment spanning the first exon (158 bp) of *eIF4E1-S*. PCR was performed in a 10-μL volume containing 20 ng/μL genomic DNA; 1X PCR buffer; 200 μM of each dNTPs; 0.1 μM of each primer; and 0.2 U *Taq* DNA polymerase. The PCR thermocycling conditions were as follows: initial denaturation at 95 °C for 10 s, followed by 7 cycles at 94 °C for 30 s and 62 °C for 30 s (decreasing the temperature at a rate of 1 °C per cycle), 40 cycles at 94 °C for 30 s, 57 °C for 60 s, and 72 °C for 90 s, and a final extension at 72 °C for 5 min. PCR products were stored at 4 °C.

### 2.4. Digestion with CEL1 endonuclease and Capillary Electrophoresis

CEL1 endonuclease purification and PCR amplification were carried out, as described previously [23]. Mutations in PCR products were detected by TILLING using the platform established by Gao et al. [23], with slight modifications. Briefly, the PCR products were denatured and reannealed. Then, CEL1 treatment was conducted in a 10-μL volume containing 2 μL of denatured-reannealed PCR products, 1.2 μL of 10X buffer, and 0.2 μL of CEL1 endonuclease. The CEL1-digested PCR products were separated by capillary electrophoresis on an AdvanCE[TM] FS96 capillary electrophoresis (Advanced Analytical Technologies, Inc., Ames, IA, USA) at 9 kV, with pre-run for 30 s, sample injection for 30 s, and sample separation for 30 min. The results of capillary electrophoresis were analyzed using the PROSize[TM] 2.0 data analysis software (Advanced Analytical Technologies, Inc., Ames, IA, USA). DNA fragments of 35 and 5000 bp were used as ladders.

### 2.5. Validation of eIF4E1-S Mutants Selected in the M3 Population

Genomic DNA was extracted from M3 *eIF4E1-S* plants, as described above, and used as a template for PCR amplification. The 489-bp amplicons were purified, and a 4-μL aliquot was used in the ligation reaction with 1 μL of pCR-Blunt II-TOPO plasmid vector (Invitrogen, Paisley, UK), according to the manufacturer's instructions. The resulting plasmid was introduced into chemically competent *Escherichia coli* cells using the heat shock method, and positive colonies were cultured overnight in LB medium containing ampicillin. Plasmid DNA was purified using the Qiagen Plasmid Mini Purification Kit (Qiagen Inc., Chatsworth, CA, USA), and the nucleotide sequence was validated by Sanger sequencing.

### 2.6. Virus Inoculation and Detection

Potyvirus inoculation was performed, as described previously [8]. Briefly, PVY[MN], PVY[O], PVY[N], and PVY[NTN] strains were propagated in burley tobacco (*N. tabacum* cv. Samsun NN) plants 30 days prior to inoculation, and the inoculum of each strain was screened using colloidal gold test strips to avoid contamination by *Tobacco mosaic virus* (TMV) or *Cucumber mosaic virus* (CMV). Leaves infected with PVY strains were homogenized in PVY inoculation buffer (0.01 M phosphate-buffered saline [PBS; pH 7] and 0.4% of $Na_2HPO_4$) using a mortar and pestle. Approximately 1% (*w/v*) of quartz sand (200 mesh) was added to the inoculum, and the mixture was filtered through a double-layered 40 mesh nylon net. Tobacco plants with 7–8 leaves were used for inoculation. The filtered inoculum was applied to two leaves per plant using a high-pressure spray gun (fluid pressure = 1 kg/cm$^2$). Plants were analyzed at 2 or 3 weeks post-inoculation by double antibody sandwich (DAS)-ELISA (Agdia-Biofords, Evry, France), according to the manufacturer's instructions.

### 2.7. Genotyping by dCAPS Markers

M3 generation *eIF4E1-S* plants carrying a nonsense mutation in exon 1 of *eIF4E1-S* were backcrossed to 'Yunyan87'. The dCAPS markers were developed for the identification of plants homozygous for *eIF4E1-S* exon 1 nonsense mutation in the original EMS mutagenized population and backcross progeny. To exclude the interference from other members of the *eIF4E* gene family, nested PCR was performed using *eIF4E1-S* gene-specific primers (isoform_c_mF_1 and isoform_c_mR_1) and dCAPS primers (Supplementary Table S1). The dCAPS primers, designed using the dCAPS Finder 2.0 program (http://helix.wustl.edu/dcaps/dcaps.html) [24], introduced a *Bsp*HI restriction site (TCATGA) in the PCR product near the stop codon mutation. The region of the gel containing the PCR products amplified using dCAPS primers was excised with a razor blade. The DNA fragments were purified from the gel matrix using the Gel Extraction Kit (Qiagen, Dusseldorf, Germany) and digested with *Bsp*HI. Enzyme digestion was incubated at 37 °C for 2 h, and digestion products were electrophoresed in an 8% non-denaturing polyacrylamide gel followed by silver staining.

### 2.8. Pilot Experiments for Agronomic Trait Analysis

Plants of $BC_4F_3$ lines and the recurrent parent 'Yunyan87' were grown in the field at Chuxiong, Baoshan, and Qujing in the Yunnan province in 2018. The experiment was performed in a randomized complete block design, with one factor (variety/line) and three replicates (10 plants of each line per replicate). Mixed fertilizer (nitrogen/phosphorus/potassium ratio=10:30:15 at Chuxiong, 15:16:25 at Baoshan, 12:12:24 at Qujing, respectively) was applied five times throughout the growth period. Various agronomic traits of plants were measured, including plant height (before and after the removal of shoot apex), leaf length, leaf width, and internode length. Differences in these above-ground traits between $BC_4F_3$ and 'Yunyan87' plants were statistically analyzed using the paired Student's *t*-test.

## 3. Results

### 3.1. Characterization of the eIF4E1-S Mutants Selected in the M2 Generation

Mutations in exon 1 of the *eIF4E1-S* gene were identified by TILLING using capillary electrophoresis (Supplementary Figure S1) and subsequently validated by sequencing (Table 1).

**Table 1.** Functional effect of amino acid substitution predicted by the PROVEAN web server.

| Mutant Line (M2) | Nucleic Acid Variation [1] | Amino Acid Variation | PROVEAN Score | Prediction | Target Region |
|---|---|---|---|---|---|
| 844 | **G**AA/**A**AA | Glu/Lys, E4K | −0.801 | Neutral | exon |
| 1001 | **G**AA/**A**AA | Glu/Lys, E4K | −0.801 | Neutral | exon |
| 113 | **G**AA/**A**AA | Glu/Lys, E4K | −0.801 | Neutral | exon |
| 139 | | | | | intron |
| 815 | A**G**A/**AA**A | Arg/Lys, R62K | 0.9 | Neutral | exon |
| 814 | A**G**A/**AA**A | Arg/Lys, R62K | 0.9 | Neutral | exon |
| 939 | **C**GG/**T**GG | Arg/Trp, R9W | −1.587 | Neutral | exon |
| 962 | | | | | intron |
| 911 | **G**AT/**A**AT | Asp/Asn, D81N | −3.302 | Deleterious | exon |
| 918 | TG**G**/TG**A** | STOP | | | exon |
| 1381 | **G**AG/**A**AG | Glu/Lys, E3K | −1.256 | Neutral | exon |
| 278 | **T**CC/**T**TC | Ser/Phe, S77F | −3.942 | Deleterious | exon |
| 1500 | **G**AA/**A**AA | Glu/Lys, E80K | −3.679 | Deleterious | exon |

[1] Bold capital letters indicate the nucleotide changes from G/C to A/T.

While two mutant lines (139 and 962) contained point mutations in *eIF4E1-S* introns, eleven mutant lines (844, 1001, 113, 815, 814, 939, 911, 918, 1381, 278, and 1500) contained point mutations in exon 1 of *eIF4E1-S*, leading to amino acid substitutions. The effect of these amino acid substitutions was predicted using the PROVEAN web server (http://provean.jcvi.org/seq_submit.php). The mutant lines 911, 278, and 1500 showed G/C to A/T mutations in exon 1, which were predicted to cause deleterious amino acid changes. Mutation in line 918 introduced a premature stop codon, resulting in a C-terminal truncated eIF4E1-S protein of only 49 amino acids. Line 918 was renamed as 918P, and plants were selfed to produce M3 seeds.

The M3 progeny of 918P was genotyped to identify *eIF4E1-S* mutants homozygous for the nonsense mutation (Figure 1A). Sequencing results confirmed that the nonsense mutation in exon 1 of *eIF4E1-S* was inherited by the M3 generation plants (Figure 1B). M3 plants homozygous for the nonsense mutation were selected and examined further for PVY resistance.

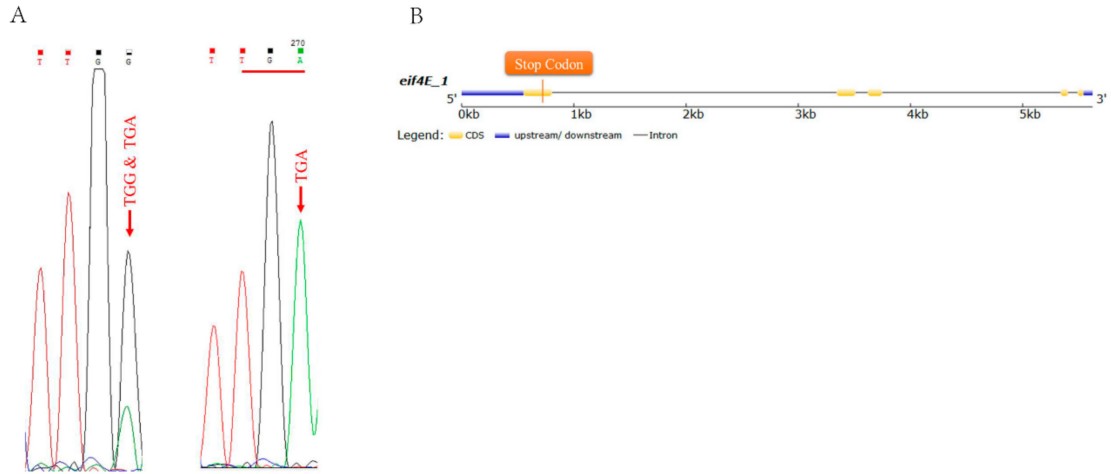

**Figure 1.** Genotyping of M3 generation progeny of line 918P: (**A**) Sequencing trace results showed mutant alleles with heterozygous genotypes (left) and mutant allele with homozygous genotype (right); (**B**) Schematic diagram of *eIF4E1-S* gene. Stop codon introduced by EMS (ethyl methanesulfonate) mutagenesis was found in exon 1 of the *eIF4E1-S* gene.

*3.2. M3 Progeny of Line 918P Exhibit PVY Resistance*

The recurrent parent 'Yunyan87', PVY resistant cultivar 'TN86' [25], and M3 generation plants were inoculated with PVY[MN] strain (Figure 2). M3 plants of line 918P homozygous for the nonsense mutation showed no disease symptoms and ELISA optical density (OD) values ranging from 0.087–0.094 (Supplementary Table S2), similar to the negative control 'TN86'. By contrast, 'Yunyan87' plants showed positive signals in ELISA with OD values ranging from 1.831–2.148. Moreover, 'Yunyan87' plants exhibited stunted growth and leaf chlorosis with a mosaic pattern (Figure 2), consistent with ELISA results.

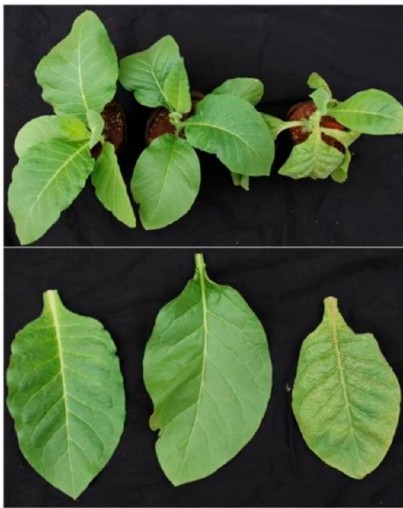

**Figure 2.** Phenotypes of wild-type 'Yunyan87', PVY (potato virus Y) resistant cultivar 'TN86', and M3 progeny of line 918P 23 days post PVY viral inoculation. Top panel from left to right: PVY resistant cultivar 'TN86', M3 progeny of line 918P, wild-type 'Yunyan87'; bottom panel from left to right: leaf sample of PVY resistant cultivar 'TN86', leaf sample of M3 progeny of line 918P, leaf sample of wild-type 'Yunyan87'.

### 3.3. Selection of Backcross Progenies using dCAPS Markers

Homozygous plants of line 918P were backcrossed to 'Yunyan87' for multiple generations to eliminate the co-inheritance of undesirable phenotypes associated with the disruption of the *eIF4E1-S* gene, and backcross progenies harboring the nonsense mutation in exon 1 of *eIF4E1-S* were selected using dCAPS markers. The dCAPS primers amplified a 327-bp fragment with a *Bsp*HI restriction site near the *eIF4E1-S* mutant allele. Digestion of the PCR products with *Bsp*HI did not affect the wild-type *eIF4E1-S* allele (because of the absence of the corresponding restriction site) but cleaved the mutant allele into two fragments (300 and 27 bp). The results showed a 300-bp band in homozygous mutants, a 327-bp band (resistant to digestion with *Bsp*H1) in homozygous wild type, and both 327- and 300-bp bands in heterozygous plants (Figure 3). The 27-bp bands in homozygous mutants and heterozygous plants ran off the gel and were not shown in Figure 3. Plants identified as homozygous and heterozygous mutants were confirmed by Sanger sequencing. This showed that dCAPS makers could be used to select backcross progeny homozygous for a specific mutation.

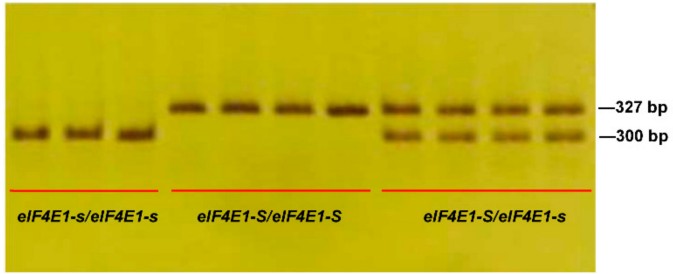

**Figure 3.** Detection of *eIF4E1-S* mutagenesis by dCAPS (derived cleaved amplified polymorphic sequence) markers. *eIF4E1-s/eIF4E1-s*, homozygous mutants with mutagenized *eIF4E1-S* allele; *eIF4E1-S/eIF4E1-S*, plants with un-mutagenized *eIF4E1-S* allele; *eIF4E1-S/eIF4E1-s*, heterozygous mutants with mutagenized and un-mutagenized *eIF4E1-S* alleles.

The selected backcross progeny was self-pollinated to produce BC$_4$F$_3$ seeds, and BC$_4$F$_3$ lines were grown together with 'Yunyan87' (control) in field trials at Baoshan, Qujing, and Chuxiong in the Yunnan province of China. Several agronomic traits, including plant height (before and after the

removal of shoot apex), leaf length, leaf width, and internode length, were measured. The $BC_4F_3$ line showed normal phenotype comparable to the 'Yunyan87' plant (Figure 4A), with no significant differences in these agronomic traits at all three field locations (Figure 4B), suggesting that *eIF4E1-S* did not affect agronomic traits and was not essential for growth in tobacco.

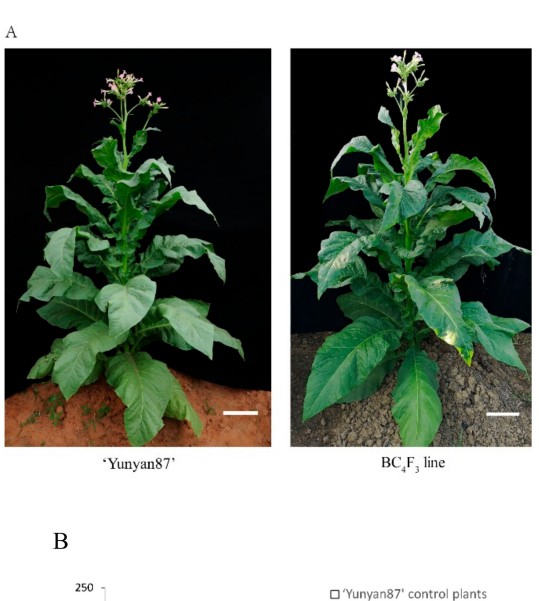

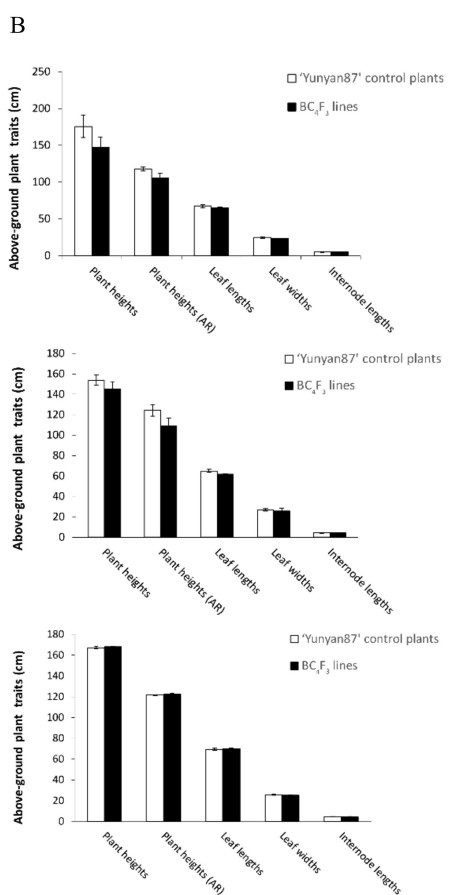

**Figure 4.** Phenotypes of 'Yunyan87' control plant and $BC_4F_3$ line. (**A**) Flowering stages of 'Yunyan87' control plant (left) and $BC_4F_3$ line (right). Bars = 100 mm. (**B**) Comparison of the above-ground plant traits between 'Yunyan87' control plants and $BC_4F_3$ lines at Baoshan (top), Chuxiong (middle), and Qujing (bottom) in the Yunnan province. Plant heights (AR): the plant heights after removal of the shoot apex. Error bars represent standard deviations (SD). Data are shown as mean ± SD (n = 3). Statistical analyses were performed using the paired Student's *t*-test.

To test the PVY resistance phenotypes of $BC_4F_3$ lines, leaf samples of $BC_4F_3$ plants, 'Yunyan87', and 'TN86' were inoculated with $PVY^O$, $PVY^N$, and $PVY^{NTN}$ strains. ELISA was performed at 14 and 21 days post-inoculation (Supplementary Table S3). Results of ELISA showed that $BC_4F_3$ plants could resist infection by all three strains, similar to the PVY resistant cultivar 'TN86', whereas 'Yunyan87' plants showed high OD values (>2.5000) (Supplementary Table S3). In the meanwhile, no disease symptoms were observed in either $BC_4F_3$ lines or 'TN86' plants at 14 and 21 days post-inoculation. These results showed that $BC_4F_3$ lines exhibited resistance to PVY isolates.

## 4. Discussion

Modification of *eIF4E* or *eIF(iso)4E* using molecular genetic approaches, such as gene knockdown by RNA interference or gene knockout using clustered regularly interspaced palindromic repeat (CRISPR)/CRISPR-associated protein 9 (Cas9), has been shown to trigger resistance against potyviruses in tomato [26], melon (*Cucumis melo*) [27], cucumber (*Cucumis sativus*) [28], and plum (*Prunus domestica* L.) [29]. Ectopic expression of the *pvr1* recessive gene in tomato [30] and overexpression of the mutant *eIF4E* gene in potato [31] confer differential resistance response against PVY strains. However, genetic transformation is strictly forbidden in tobacco breeding, and the regulatory fate of CRISPR/Cas9 for use in agriculture remains unclear in some countries [32]. Thus, mutagenesis of *eIF(iso)4E* genes using EMS is the preferred approach to generate resistant alleles. The feasibility of this approach has been demonstrated in *N. tabacum*, where stop codon mutations (W50* and W53*) have been introduced in exon 1 of *eIF4E1-S* in the PVY susceptible burley tobacco line BB16NN [22]. Both of these EMS-induced nonsense mutations lead to the synthesis of C-terminal truncated eIF4E1-S proteins, which confer resistance to $PVY^O$ and $PVY^N$ isolates [22,33] but reduced resistance to PVY isolates of the C clade [33].

In this study, the EMS-mutagenized M2 population of flue-cured tobacco 'Yunyan87' was screened by TILLING to identify mutants harboring a mutation in exon 1 of the *eIF4E1-S* gene. The nonsense mutation in *eIF4E1-S* was inherited by the subsequent M3 generation, and M3 plants homozygous for this mutation showed resistance to the $PVY^{MN}$ isolate (potato tuber necrotic ringspot strain of PVY belonging to the C clade). These results were in accordance with previous findings, showing the direct involvement of *eIF4E1-S* in the recessive resistance against potyvirus [22].

Frequent emergence of resistance-breaking (RB) PVY variants has been associated with EMS mutagenized *eIF4E1-S* knockout mutants [22]. Tolerance to RB PVY variants varies with the virus isolates [34–37] and mutation types affecting *eIF4E* genes, such as *eIF(iso)4E-T* and *eIF(iso)4E-S* [38]. A recent study characterized the differences in resistance durability, in which mutants harboring a large deletion at the *eIF4E1-S* locus on chromosome 21 displayed the most durable resistance, whereas those carrying frameshift and nonsense mutations displayed less durable resistance and were associated with frequent emergence of RB PVY isolates [33]. Additionally, genetic and transcriptomic analyses conducted by Michel et al. [33] indicated that resistance durability was correlated with a complex genetic locus on chromosome 14 that contained three other *eIF4E* copies (*eIF4E-2–4*). However, in this study, we did not observe the breakage of resistance against PVY isolates, possibly because a limited number of isolates were assayed. To determine the resistance durability of the 918P homozygous mutants and the involvement of other *eIF4E* copies in PVY resistance, the integrity of *eIF4E-3* and expression levels of *eIF4E-2* in 918P homozygous mutants would need to be characterized further.

Altogether, the dCAPS markers developed in this study were used for the identification of plants homozygous for *eIF4E1-S* exon 1 nonsense mutation in the original EMS mutagenized population and backcross progeny. The obtained $BC_4F_3$ plants not only showed resistance against different potyvirus strains but also showed normal phenotypes comparable to parental genotype 'Yunyan87'. Thus, 918P homozygous mutant plants could serve as valuable germplasm in future tobacco breeding programs for improving PVY resistance of flue-cured tobacco.

## 5. Conclusions

EMS-mutagenized M2 population of 'Yunyan87' was screened by TILLING, and 'Yunyan87' mutants harboring the nonsense mutation in exon 1 of *eIF4E1-S* were identified. The M3 generation progeny homozygous for this mutation conferred resistance to PVY strains. The backcross progeny $BC_4F_3$ selected by dCAPS markers resisted the infection by PVY strains and showed normal phenotypes comparable to parental genotype 'Yunyan87'.

**Supplementary Materials:** The following are available online at http://www.mdpi.com/2073-4395/10/1/36/s1, Figure S1: Mutation detection using capillary electrophoresis, Table S1: Primers used in PCR and dCAPS assay, Table S2: ELISA of PVY resistance of selected tobacco lines, Table S3: ELISA of PVY resistance of selected tobacco plants subjected to the inoculation of different potyvirus strains.

**Author Contributions:** Conceptualization, Z.S., L.Z., W.L. and B.W.; Methodology, X.Y., X.S. and Y.G.; Software, W.L. and L.Z.; Validation, L.Z., B.W. and Z.S.; Formal Analysis, L.Z.; Investigation, Z.S., L.Z., W.L., B.W., X.S., Y.G., Y.L. and X.C.; Resources, F.J.; Writing—Original Draft Preparation, L.Z.; Writing—Review and Editing, L.Z., X.Y., X.S. and B.W.; Supervision, Z.S. All authors have read and agreed to the published version of the manuscript.

**Funding:** This work was supported by the Yunnan Academy of Tobacco Agricultural Sciences (Grant No. 2012YN02, 2017YN02, 2017YN03).

**Conflicts of Interest:** The authors declare no conflict of interest.

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
