# Peer review of "Development of a PVY Resistant Flue-Cured Tobacco Line via EMS Mutagenesis of eIF4E"

_agronomy, doi:10.3390/agronomy10010036_

Round 1
Reviewer 1 Report
Experiments are well planned and their results are presented and discussed nicely
Author Response
Thank you very much for your comments.Reviewer 2 Report
This manuscript describes the generation of a EMS-TILLING population of flue-cured tobacco and the identification of mutations in the eIF4E1-S gene. A nonsense mutation in the first exon of the eIF4E1-S gene is selected und homozygous plants shown to be resistant to several PVY isolates (-MN/O/N/NTN). A screening method using derived cleaved amplified polymorphic sequence (dCAPS) markers is described.
The manuscript is well written, experiments correctly performed. The correlation of eIF4E and PVY resistance has been described before, but the authors claim that the created mutant could be useful for tobacco improvement. Durability of this resistance could maybe be added to this work to improve its interest.
Minor comments:
Page 2 line 48: genome-linkED
Line 175: “The mutant lines 911, 278, and 1500 showed G/C to A/T mutationS in exon 1, which WERE predicted to cause deleterious amino acid CHANGES.
Table 1: glu&lys: anino acid in three letter code start with uppercase-> Glu or Lys. I would use “/” instead of “&”, thus Glu/Lys or write “Glu to Lys”
Line 202: “Homozygous plants of line 918P were backcrossed to ‘Yunyan87’ for multiple generations, …” and later “Plants with the mutagenized eIF4E1-S allele were crossed with ‘Yunyan87’, and the co-inheritance of undesirable phenotypes associated with the disruption of the eIF4E1-S gene were eliminated though subsequent backcrossing.” the action described in these two sentences is the same, paragraphs should be rewritten.
Line 222. No significant morphological differences were observed between BC4F3 lines and ‘Yunyan87’ plants….” And line 226 “ No significant differences were…” here also the sentences are duplicated and should be fused. Morphological differences include the agronomical traits.
Tables with Elisa results can be supplemental data, giving average of values for positive and negative infection in the text.
The first paragraph of discussion is a repetition of the introduction and can be significantly shortened.
Reviewer 3 Report
This is a very nice study, has high significance in tobacco breeding in future against PVY and is well-written. I have only few minor comments below-
1. Line 237: Apart from doing ELIZA test, did you also visually observed and rated the plants for PVY symptoms. If so, presenting the disease rating data might be useful.
2. Line 286: What about its potential effect on yield? Any trade-off expected for these PVY resistant plants?
Thank you.
